A systematic review of automated pre-processing, feature extraction and classification of cardiotocography

Al-yousif Shahad shahad.alyousif@duc.edu.iq 1 2
Jaenul Ariep 3
Al-Dayyeni Wisam 1
Alamoodi Ah 4
Najm IA 5
Md Tahir Nooritawati 6
Alrawi Ali Amer Ahmed 7
Cömert Zafer 8
Al-shareefi Nael A. 9
Saleh Abbadullah H. 10
1 Department of Medical Instrumentations Engineering Techniques, Dijlah University , Baghdad , Iraq
2 Faculty of Information Science & Engineering, Management and Science University , Shah Alam , Selangoor , Malaysia
3 Department of Electrical Engineering, Faculty of Engineering and Computer Science, Jakarta Global University , Jakarta , Indonesia
4 Department of Computing, Universiti Pendidikan Sultan Idris , Tanjong Malim , Perak , Malaysia
5 Faculty of Engineering, Tikrit University , Tikrit , Iraq
6 Faculty of Electrical Engineering, Universiti Teknologi MARA (UiTM) , Shah Alam , Selangor , Malaysia
7 Training Directorate, Ministry of Science and Technology , Baghdad , Aljadireyah , Iraq
8 Department of Software Engineering, Samsun University , Samsun , Turkey
9 College of Biomedical Informatics, University of Information Technology and Communications (UOITC) , Baghdad , Almansoor , Iraq
10 Department Computer Engineering, Karabük University, , Karabük , Karabük , Turkey
Issac Biju
Electronic publication date: 2021 Apr 27
Publication date: 2021
Volume: 7
Electronic Location ID: e452
Received 2020 Oct 20; Accepted 2021 Mar 1
Copyright: ©2021 Al-yousif et al.
Copyright year: 2021
Copyright holder: Al-yousif et al.
License: This is an open access article distributed under the terms of the Creative Commons Attribution License, which permits unrestricted use, distribution, reproduction and adaptation in any medium and for any purpose provided that it is properly attributed. For attribution, the original author(s), title, publication source (PeerJ Computer Science) and either DOI or URL of the article must be cited.
License URL: https://creativecommons.org/licenses/by/4.0/

Keywords: Cardiotocography, Classification, Fetal Heart Rate, Baseline, Variability, Feature extraction, Uterine contraction, Diagnoses, Acceleration

Funding: The authors received no funding for this work from any other public or private institutions; we self funded this work. The funders had no role in study design, data collection and analysis, decision to publish, or preparation of the manuscript.

==============================
Context

The interpretations of cardiotocography (CTG) tracings are indeed vital to monitor fetal well-being both during pregnancy and childbirth. Currently, many studies are focusing on feature extraction and CTG classification using computer vision approach in determining the most accurate diagnosis as well as monitoring the fetal well-being during pregnancy. Additionally, a fetal monitoring system would be able to perform detection and precise quantification of fetal heart rate patterns.

Objective

This study aimed to perform a systematic review to describe the achievements made by the researchers, summarizing findings that have been found by previous researchers in feature extraction and CTG classification, to determine criteria and evaluation methods to the taxonomies of the proposed literature in the CTG field and to distinguish aspects from relevant research in the field of CTG.

Methods

Article search was done systematically using three databases: IEEE Xplore digital library, Science Direct, and Web of Science over a period of 5 years. The literature in the medical sciences and engineering was included in the search selection to provide a broader understanding for researchers.

Results

After screening 372 articles, and based on our protocol of exclusion and inclusion criteria, for the final set of articles, 50 articles were obtained. The research literature taxonomy was divided into four stages. The first stage discussed the proposed method which presented steps and algorithms in the pre-processing stage, feature extraction and classification as well as their use in CTG (20/50 papers). The second stage included the development of a system specifically on automatic feature extraction and CTG classification (7/50 papers). The third stage consisted of reviews and survey articles on automatic feature extraction and CTG classification (3/50 papers). The last stage discussed evaluation and comparative studies to determine the best method for extracting and classifying features with comparisons based on a set of criteria (20/50 articles).

Discussion

This study focused more on literature compared to techniques or methods. Also, this study conducts research and identification of various types of datasets used in surveys from publicly available, private, and commercial datasets. To analyze the results, researchers evaluated independent datasets using different techniques.

Conclusions

This systematic review contributes to understand and have insight into the relevant research in the field of CTG by surveying and classifying pertinent research efforts. This review will help to address the current research opportunities, problems and challenges, motivations, recommendations related to feature extraction and CTG classification, as well as the measurement of various performance and various data sets used by other researchers.

Introduction

Every mother wants a healthy pregnancy, normal birth pregnancy and a healthy baby. This condition must be supported with regular prenatal care. For a mother, examinations are essential for detecting if there are any problems in pregnancy, preparing physically and mentally, knowing the condition of the pregnancy, and to determine the method of delivery that is in accordance with the results of the test. The purpose of CTG recordings is to identify when there is concern about fetal well-being to allow interventions to be carried out before the fetus is harmed. The focus is on identifying fetal heart rate (FHR) patterns associated with inadequate oxygen supply to the fetus. As for fetal testing is important to ensure the exact condition of the fetal so that high-risk births could be minimized, and Cardiotocography is one way to conduct prenatal testing (Chamidah & Wasito, 2015).

Generally, cardiotocography is a practical method that can be used to examine fetal well-being and at the same time widely used for intrapartum and antepartum fetal monitoring (Nagendra et al., 2017; Zhang & Zhao, 2017; Das, Roy & Saha, 2015b). Cardiotocography (CTG) is also called Electronic Fetal Monitoring (EFM), EFM was first introduced around 1960, and became the first tool to use phonocardiography to record Fetal Heart Rate (FHR), and was then replaced by a doppler signal with significant improvements to signal quality (Pinas & Chandraharan, 2016). In addition, cardiotocography is the main biophysical method for monitoring the condition of the fetus during pregnancy and childbirth. CTG consists of non-invasive recording (using the doppler ultrasound technique) from changes in FHR and analyzes the relationship between fetal movement and maternal uterine contractions (Haweel & Bangash, 2013). Information obtained from CTG is further used as an initial identification of pathological conditions (for instance fetal suffering, congenital heart deficiency, hypoxia and others) and can assist doctors to anticipate further for any complications and early permanent damage to the fetus (Sahin & Subasi, 2015).

From the recorded data attained using CTG, the obstetrician can extract and evaluate four parameters which form the basis of CTG feature extraction based on international medical guidelines (Tomas et al., 2013; Shah et al., 2015). These parameters are:

• Baseline - represents the physiological value with a range of 110–160 b.p.m, and if the baseline is more than 160 b.p.m, the baseline is called tachycardia, whereas if the baseline is below 110 b.p.m, the baseline is called bradycardia.

• Acceleration increases in the baseline higher than 15 b.p.m for at least 15 s. In 15 min, the acceleration must occur at least twice. At night acceleration is considered a pathological state and can also be considered as a response to fetal movement.

• Deceleration - Decrease in the baseline of more than 15 b.p.m at least 15 s. If deceleration is perceived with contraction, possibility indicates with hypoxia.

• Variability - Fluctuation of the baseline and is not considered as acceleration or deceleration. There are differences between variability during sleep state and perform activities or in active state.

Conversely, FHR signal monitoring is carried out by the obstetrician using eye inspection during the critical labor period to assess the condition of the fetus (Frigo & Giorgi, 2017). However, the interpretations made by humans are inconsistent namely about the traces and variability of inter and intra high observers (Cömert & Kocamaz, 2017b; Jyothi, Hiwale & Bhat, 2016; Gavrilis, Nikolakopoulos & Georgoulas, 2015; Fergus, Selvaraj & Chalmers, 2018). The interpreted results are subjective and cannot be reproduced. This is the starting point for the debate on the effectiveness of cardiotocography in gynecology work or not, particularly in the cases of low-risk pregnancies. In addition, this results is too dependent on the test which may lead to an increase in diagnosis errors from fetal distress and further increases the number of cesarean deliveries (Permanasari & Nurlayli, 2017). Despite the development in CTG guidelines to evaluate the recording of cardiotocography, a misdiagnosis still occurs because of the difference in experience and tacit knowledge among obstetricians is a big dilemma (Ocak, 2013).

To increase the assessment objectivity and repeatability, the computer-aided fetal monitoring systems provide an automated quantitative analysis of the signals. It allows for accurate detection and precise quantification of the FHR patterns and provides additional information that is invisible to the naked eye (Czabański et al., 2013; Czabanski et al., 2015). Analysis and classification of cardiotocography using computerized systems are widely applied to increase the effectiveness and utility of CTG monitoring and also reducing inconsistencies in interpretation, many efforts have been made by researchers from medical and technical backgrounds (Ocak & Ertunc, 2013).

Our research aims to describe the achievements made by the researchers, summarizing findings that have been found by previous researchers in preprocessing, feature extraction and cardiotocography classification, to determine assessment methods and criteria, to suggest literature taxonomies in the CTG field and to distinguish aspects from relevant research in the field of cardiotocography. The preparation in this study was carried out in this way: ‘Introduction’ introduces the field of research that we are doing; ‘Systematic Review Sources’ describes the protocol of the proposed systematic review; the taxonomy style is presented in ‘Methods’; statistical results of the articles reviewed is elaborated in ‘Discussion’; discusses guidelines, data sets, validation techniques, and measures of performance, motivation, challenges, and recommendations taken from articles reviewed; ‘Limitation’ discussed limitations of the study; and in ‘Conclusion’ findings from this reviews are concluded.

Systematic Review Sources

Database

Article search is done systematically using three databases such as (1) IEEE Xplore digital library, (2) Science Direct and (3) Web of Science (WOS). Article searches are based on the facilities provided by the three databases both in simple and complex searches by searching for various journals and conference articles in various fields of science such as engineering, computer science and medical. Therefore, the literature in the medical sciences and engineering is included in the search scope to provide broader understanding and views for researchers related to a variety of science majors.

Article selection procedure

The selection procedure of the article in this study was based on an intensive search for the relevant publications in this field of study using the following steps: (1) Scan titles and abstracts that are in accordance with the topic and exclude inappropriate and not related papers and to find any duplication of papers. (2) The next step is to read the full papers thoroughly.

Article search

The articles search was conducted using three databases namely IEEE, Science Direct (SD) and Web of Science (WoS) through the database search on March 17, 2018. In all three databases, searches were performed using queries (Cardiotocogram OR Cardiotocograph OR Cardiotocography) with the following keywords (“fetal heart rate” OR baseline OR “baseline variability” OR acceleration OR deceleration OR “uterine contractions”). Searches are added using this keyword (“FHR-monitoring” OR “feature extraction” OR classification OR diagnosis) to limit the search space to cardiotocography cases related to engineering and computer science. Search queries are shown in Fig. 1.

Figure 1 Article selection, search query and inclusion criteria.

Article searches are added with advanced search options for each database used to select conference and journals articles, and excluded articles from each chapter of the book or other documents. Articles in journals and conference articles are chosen for consideration because articles in journals and conferences are most likely involved in the latest research and relevant to this study. The presentation of rule information used in conducting search requests is shown in Table 1.

Table 1 Search query settings.

Sources	IEEE	Science Direct	Web of Science	
Year s	2013–2018	2013–2018	2013–2018	
Languange	English	English	English	
Run on	Full Text	Full Text	Full Text	
Subject areas	All Available	All Available	All Available	
Date of running search	17/03/2018	17/03/2018	20/03/2018	

Criteria for selected articles

Figure 1 shows the criteria for the selected article and all articles that meet the criteria are selected. Mapping of research space on feature extraction and cardiotocography classification becomes a general taxonomy and a rough overview of the four categories defined as initial targets. The division of this category is based on an intensive pre-survey of the selected literature resources. Any article that did not match the specified criteria as depicted in Fig. 1 was not selected. Criteria that are not included are as follows: (1) Papers not written in English. (2) Focus on fetal heart rate and uterine contraction extraction and classification but with manual analysis features.

The process of collecting articles

In managing the list of articles, the author uses the Excel application which is composed of the three databases used. The selected article is read in full by the authors and given several highlights and important comments, and the taxonomy that has been set at the beginning is refined in the classification of the articles in the listing. Each article is Summarize, tabulate, and all-important findings are elaborated in detail. The word and excel application are used to store related information such as lists of surveyed articles, source indexes, summary tables and descriptions, objectives, review sources, audiences, data sets used, criteria of evaluation, techniques of validation, etc. These data are presented in the supplementary materials as a comprehensive indication of the obtained results.

Article statistics information and search results article

The taxonomy used to review these research articles that focus on automatic feature extraction and cardiotocography classification is as depicted in Fig. 2. This taxonomy stated the entire development of various researches and modern applications. The classification stated four main stages. The method proposed is discussed in the first stage (stage 1), this includes all proposed method that presents steps and algorithms in the preprocessing stage, feature extraction and classification and their use in cardiotocography (20/50 articles). Development of system for automatic feature extraction along with cardiotocography classification is conducted in second stage (stage 2) (7/50 articles). Next, the third stage (stage 3) comprised of reviews and survey articles related to automatic feature extraction and cardiotocography classification (3/50 articles). Finally, evaluation as well as comparative studies in determining the most optimum technique related to extracting and classifying features and its comparisons is discussed in the last stage (stage 4) (20/50 articles).

Figure 2 The research literature taxonomy on automatic feature extraction and classification of cardiotocography.

Methods

As shown in Fig. 2, in this review the categorization of the CTG methodology is projected into four stages.

Techniques and algorithms

This section discusses the methods and algorithms used in feature extraction and cardiotocography classification of all articles that have been included in this study. The general context of articles in this taxonomy is about achieving high performance or increasing the accuracy of cardiotocography. This first stage has 20 articles out of 50 and it is divided as follows:

• The first subclass {7/20 articles}: focuses on the classification using the new proposed method or modify conventional methods.

• The second subclass {4/20 articles}: focuses on preprocessing and classification methods.

• The third subclass {6/20 articles}: discusses feature extraction (features selection) and classification methods.

• The fourth subclass {3/20 articles}: discusses preprocessing, feature extraction and classification methods.

As mentioned earlier, the first segmented group of articles discussed classification. Firstly, Volterra based Neural Networks (VNN) using the Volterra series expansion are proposed in Haweel & Bangash (2013). Secondly, the scattering transform is proposed in Chudáček et al. (2014) as a new method for analyzing the variability of FHR in the intrapartum period. Thirdly, The CTG trail using a 3-level classification presented in Das, Roy & Saha (2015a) to classify CTG as Normal (N), Suspicious (S), and Pathological (P). Further, an ordinal classification approach as used in Georgoulas et al. (2017) based on Binary Decomposition for prediction, that used C.45 algorithm split the data and select a test that provided the best information gain, and used Synthetic Minority Oversampling Technique (SMOTE). Next, Permanasari & Nurlayli (2017) proposed a method that implemented in the decision tree (DT) to analyze the CTG data to determine Fetal Distress. Moreover, in clinical practice, the qualitative assessment of FHR records used fuzzy classification with the Weighted Fuzzy Scoring System (WFSS) technique as proposed in Czabański et al. (2013). Lastly, proposed a scheme that is based on adaptive neuro-fuzzy inference systems (ANFIS) was trained to predict the normal and the pathological state (Ocak & Ertunc, 2013).

Next, the second segmented group discussed preprocessing and classification. Firstly, in Georgoulas et al. (2014) Interpolate using Matlab implementation of Hermite Spline Interpolation proposed a classification approach used for missing data, the technique is based on a latent class analysis method that seeks to produce more objective labeling of training cases, and this is an important step in any classification problem. Secondly, as proposed in Gavrilis, Nikolakopoulos & Georgoulas (2015) Principal Component Analysis (PCA) used both as part of the one class classification process as well as for visualization, while Isomap, a nonlinear dimensionality method, is used to confirm the visualization results of the PCA and in this study, two simple one-class classifiers are tested: the nearest neighbor (NN) and the Parzen window classifier. Thirdly, preprocessing proposed in Jyothi, Hiwale & Bhat (2016) used a moving average filter to compensate for the noise and for contraction classification using K-Nearest Neighbor (KNN). Lastly, in preprocessing using Principal Component Analysis (PCA), the data utilized for training and testing using the Adaptive Boosting algorithm (AdaBoost) to obtain strong groupings for classifying unknown data and predict the state of the fetus proposed in Zhang & Zhao (2017).

Further, the third group discussed feature extraction and classification. Firstly, in Shah et al. (2015), Shah et al. proposed the use of correlation feature selection subset evaluation (CFS) with CFS calculated the correlation between features and classes for classification, Random Forest applied to classify data samples into classes and REPTree applied to produce decision trees, by calculating information acquisition using entropy and also using J48. Secondly, the K-means algorithm utilized to obtain the important features and for classification Support Vector Machine (SVM) was used as reported in Chamidah & Wasito (2015). Thirdly, as proposed in Ocak (2013), the Genetic Algorithm (GA) implemented to minimize the number of features and determine the ideal subset of features and further classify fetuses using Support Vector Machine (SVM). Fourthly, feature extraction based on Genetic Algorithm (GA) used as proposed in Xu et al. (2014) and as for classification linear regression, linear Support Vector Machine and kernel radial basis function (RBF) SVM were utilized. Moreover, as presented in Inbarani, Banu & Azar (2014), in order to identify the most important features, unsupervised swarm-based reduction techniques, hybrids of swarm intelligence and rough sets were employed and to check the level of errors generated from the reduce set by applying K-means clustering, and for classification, this study used single decision tree (DT), multilayer perceptron (MLP) neural network, probabilistic neural network (PNN) and random forest. Finally, as proposed in Kim, Yang & Lee (2017), it used fitMine as a new nonlinear dynamic model that reflected the relationship between signals of Fetal Heart Rate and Uterine Contraction by combining chaotic population model and unscented Kalman filter algorithm. In the last segmented group of articles, it discussed three components namely the preprocessing method, feature extraction, and classification. Firstly, as reported in Tomas et al. (2013) were used the principal component analysis (PCA) used as feature reduction, the correlation feature selection subset evaluation (CFS), CFS calculated the correlation between features and classes and three algorithms were used in this work: Random Forest, REPTree and the Linear Discriminant Analysis. Secondly, for preprocessing was used R-R intervals , while sequential feature selection algorithm was used to determine the informative subset of HRV features and Support Vector Machine was used to classify healthy fetuses and fetuses with adverse outcome presented in Warmerdam et al. (2016). Lastly, (Spilka et al., 2017) proposed cubic spline interpolation for preprocessing, as for feature extraction this study used FIGO enhanced and automated features (FI), Spectral Energy (SC) and Multiscale multifractal analysis (SI) and for classification data Sparse Support Vector Machine (Sparse-SVM) was utilized.

System development

This section discusses either a full-automatic or semi-automatic system that is used for feature extraction and CTG classification for system development. This second stage has 7 articles out of 50 and it is divided as follows:

• The first subclass {5/7 articles}: based on supervised classification engine.

• The second subclass {2/7 articles}: based on unsupervised classification engine.

The first subclass discusses the phase of the system. The following method is carried out in this literature: Preprocessing, Segmentation and Identification in Pasarica et al. (2017), Preprocessing, identification of parameter, clinical validation, performances indexes estimation and statistical parameters in Romano et al. (2016), preprocessing, Identifying and classify in Rotariu et al. (2014), Preprocessing and classification in Georgieva et al. (2013), Feature extraction and classification in Czabanski et al. (2015). In the second subclass, the related work focused on the proposed system in Wróbel et al. (2013) that used the weighted myriad filtering with identification and selection, and in Das, Roy & Saha (2015b) agreement between the proposed technique and visual interpretation by obstetricians was assessed using Bland-Altman analysis.

Review/survey

The third Stage includes reviews and surveys aimed at describing cardiotocography, methods and techniques used in feature extraction and cardiotocography classification. This third stage has 3 articles out of 50 and were found in Warmerdam et al. (2018); Fergus, Selvaraj & Chalmers (2018); Bhatia et al. (2017). The first in Warmerdam et al. (2018) presented Fetal HRV analysis that provided distress information on fetal, combining HRV features calculated over the entire fetal heart rate with contraction-dependent HRV features that improved the classification performance during the second stage of labor. The second in Fergus, Selvaraj & Chalmers (2018), showed a review of human cardiotocography interpretations and giving an opinion that machine learning can be used as a decision support system by obstetricians and midwives and also can provide objective results in addition to using normal practice. Finally as reported in Bhatia et al. (2017), this study presented a comparison of the cardiotocography classification system as outlined by the International Federation of Gynecology and Obstetrics (FIGO) in 2015 and the UK National Institute for Health and Care Excellence (NICE) in 2007 and 2014.

Evaluation and comparative study

This section, the fourth stage, discusses the efforts to determine the best method for feature extraction and cardiotocography classification by comparing methods based on several criteria. This fourth stage has 20 articles out of 50.

The two classification schemes were compared that depended on one of the two subclasses:

• The error rate (stage 4/ subclass 1) shown in the articles (Frigo & Giorgi, 2017; Romano et al., 2018).

• The reliability criteria group (stage 4/ subclass 2) as reported in the articles (Cömert, Kocamaz & Güngör, 2016; Magenes et al., 2016; Nagendra et al., 2017; Sahin & Subasi, 2015; Rei et al., 2015; Di Tommaso et al., 2013; Cömert & Kocamaz, 2017a; Gamboa et al., 2017; Chen et al., 2014; Arif, 2015; Garabedian et al., 2017; Pinas & Chandraharan, 2016; Ghi et al., 2016; Cömert & Kocamaz, 2017b; Yılmaz, 2016; Chinnasamy, Muthusamy & Gopal, 2013; Sundar, Chitradevi & Geetharamani, 2014; Nunes & Ayres-de Campos, 2016).

One of the contributions to this work is to prioritize the latest articles in the research literature using signal processing through content analysis from several major journals in the field of study. The number of articles obtained from the literature is shown in Fig. 3. These articles are (50) articles selected based on predetermined criteria. As depicted in Fig. 3, interest in the development of methods and systems for feature extraction and cardiotocography classification has increased accordingly over the rest.

Figure 4 shows the number of articles based on each category and publication year. A total of 20 articles explained the proposed method in the study, proposals for system development totaling of 7 articles, reviews and surveys totaling of 3 articles and evaluations and comparisons of two or more methods amounted to 20 articles.

Figure 3 Articles number based on main categories and the database.

Figure 4 The number of articles in each category based on the year of publication.

Figure 5 shows the author’s affiliation country (the main author’s country is considered in a case study with several co-authors). Majority of the studies focused on feature extraction and cardiotocography classification are in Italy, Turkey and India. Other countries include England, the Czech Republic, Poland, Indonesia, Sweden, the Netherlands, Saudi Arabia, Portugal and Romania. The number of co-authors was 145 out of 50 articles reviewed. Co-authors come from various universities in the world with the majority coming from the fields of biomedical engineering, bioinformatics, signal processing, computer science, medicine, and pharmacy.

Figure 5 Articles number based on the author’s affiliation.

Discussion

This study focused more on literature as compared to techniques or methods. This is the main difference of this study in comparison to the other reviews. The literature was organized as a proposed taxonomy. The taxonomy development of literature in a certain study area is helpful, especially in an emerging research field. On the one hand, the taxonomy of literature can regulate many publications, such as researchers can recognize the research developments and activities in the field of the research area through research papers structuring. The comprehensive taxonomies can describe the actual activities in the research area. Some articles can handle the introductory aspects, while other articles can handle the problems of techniques and methods used now, and the rest can contribute significantly to provide solutions in the field of research. Relevant taxonomic literature can help organize various articles and research into manageable, coherent and meaningful layouts Moreover, the order given by taxonomy gives the researchers useful understandings into the topic in various aspects. First, explaining the main research areas and subdirectories such as taxonomic examples on feature extraction and cardiotocography classification in this study showed that researchers in the articles are more likely to suggest methods for classification; so that, this research area can be considered as a promising path. This is also another important area is the development of feature extraction and classification for cardiotocography. Second, the taxonomy can reveal the gaps in the research area. Mapping the research on automatic techniques in extracting features and classification of cardiotocography into different classes can emphasize and highlight strength and limitation in the scope of each study. For example, the taxonomy proposed in this study showed interesting and prominent topics in terms of the methods proposed in previous studies, publication of surveys and evaluations and comparative studies. Furthermore, insights obtained from statistics on a single class of suggested taxonomies showed an active direction in feature extraction and cardiotocography classification to join the new research trends and strengthen the inactive research areas. Finally, the researchers who conduct a study in this area often recommend and adopt this taxonomy, promoting this taxonomic plan as a reference to assist in joint work and discussion with other researchers, such as articles in development, comparison of research and review of various automated methods that are not similar along with extraction feature techniques and cardiotocography classification. In this study too, research and identification of various types of datasets used in surveys are also conducted because identifying datasets used by previous studies was indeed vital. In addition, this study also presents various important aspects of evaluation based on classification methods used. Further, a review of the various validation techniques and performance evaluation criteria adopted is conducted. Surveys conducted from the content of the literature reveal six aspects: (1) The description of the datasets used in the article, (2) The validation methods adopted, (3) The techniques used for the evaluation performance of the methods, (4) The motivation to use the automatic methods in detecting and classifying cardiotocography, (5) The challenges to success in using the method and (6) The recommendations for overcoming difficulties.

This research to shed light on cardiotocography researches, which provide valuable information for researchers and specialists in in gynecology and obstetrician clinics, in addition to the engineers working in the field of medical engineering and information technology. Where by collecting all data and studies in the field of cardiotocography and discussing its précised details and highlighting the problems, obstacles, motivation, challenges, recommendations and limitations of this research will gives a clear vision for those who are interested in this field, especially academic researchers.

Datasets

In this survey, there are 3 categories of datasets discussed: publicly available, private dataset and commercial dataset. In publicly dataset, researcher can download the datasets freely, in private dataset researcher need to request the private from infirmaries, university or research Centre and for commercial dataset it can be obtained from Clinical Environment. In our survey, we got 13 sources from UCI Machine Repository, CTU-UHB Database (University Hospital), Hospital Femme Mere Enfant (HFME), CTU-UHB (Czech Technical University), Azienda Ospedaliera Universitaria Federico II, Registration for CTG is recorded at a Neoventa STAN Molndal Sweden, University Hospital of Bologna, Tertiary care University Hospital, Clinical Environment (Romano et al., 2018), Clinical Environment (Romano et al., 2016), Careggi University Hospital of Florence, Mackay Memorial Hospital, a tertiary referral center, and Maternity and Gynecological Clinic (University Hospital of Porto in Portugal).

From our survey, UCI Machine Repository is the most widely used, secondly CTU-UHB database from university hospital in Brno Czech Republic, thirdly CTU-UHB database from Czech Technical University, and lastly Hospital Femme Mere Enfant (HFME). Four of these datasets are publicly available in their website, so researchers can download the database freely. Table 2 summarizes the basic details of the cardiotocography dataset in the survey that we have conducted.

Table 2 Dataset used in reviewed research.

No.	Ref	Datasets	Type	Source	
1	 Tomas et al. (2013), Haweel & Bangash (2013), Gavrilis, Nikolakopoulos & Georgoulas (2015), Shah et al. (2015), Chamidah & Wasito (2015), Cömert, Kocamaz & Güngör (2016), Georgoulas et al. (2017), Permanasari & Nurlayli (2017), Nagendra et al. (2017), Zhang & Zhao (2017), Sahin & Subasi (2015), Ocak (2013), Ocak & Ertunc (2013), Yılmaz (2016), Chinnasamy, Muthusamy & Gopal (2013), Inbarani, Banu & Azar (2014), Sundar, Chitradevi & Geetharamani (2014)	Publicly available	Each record provides information about morphological patterns (physiological, suspect, pathological)	UCI Machine Repository
https://archive.ics.uci.edu/ml/datasets.html	
2	 Georgoulas et al. (2014), Rotariu et al. (2014), Cömert, Kocamaz & Güngör (2016), Jyothi, Hiwale & Bhat (2016), Cömert & Kocamaz (2017b), Kim, Yang & Lee (2017), Pasarica et al. (2017)	Publicly available	Consisting of 552 records obtained between 2009 and 2012	CTU-UHB database
From the midwifery ward of the university hospital in Brno, Czech Republic
https://physionet.org/physiobank/database/ctu-uhb-ctgdb/	
3	 Chudáček et al. (2014), Spilka et al. (2017)	Private Datasets	Intrapartum CTG has been routinely monitored in HFME for the past 30 years, with systematic monitoring based on STAN	Hospital Femme Mere Enfant (HFME) (Lyon, France)
http://www.chu-lyon.fr/en/hopital-femme-mere-enfant	
4	 Das, Roy & Saha (2015b), Das, Roy & Saha (2015a), Kim, Yang & Lee (2017)	Publicly Available	Normal and Pathological datasets	CTU-UHB
From Czech Technical University, Department of Cybernatics
https://physionet.org/physiobank/database/ctu-uhb-ctgdb/	
5	 Magenes et al. (2016)	Private Dataset	Normal fetuses and IUGRs	Azienda Ospedaliera Universitaria Federico II, Napoli, Italy	
6	 Warmerdam et al. (2016)	Private Dataset	22 cases were included with adverse results, which was matched with 110 healthy cases	Registration of CTG is recorded at Neoventa STAN (Molndal, Sweden)	
7	 Ghi et al. (2016)	Private Dataset	Retrospective nesting case-control studies including a series of consecutive fetuses delivered with metabolic acidemia in the second stage of labor between 2008 and 2013	University Hospital of Bologna	
8	 Rei et al. (2015)	Private Dataset	100 and 51 CTG tracings were consecutively selected from pre-existing database of intrapartum tracings.	Tertiary care University Hospital	
9	 Romano et al. (2018)	Commercial Dataset	Recorded from healthy pregnant women	Clinical Environment  Romano et al. (2018)	
10	 Romano et al. (2016)	Commercial Dataset	All CTGs are recorded during routine daily fetal monitoring in the clinical environment of women between 31 and 41 weeks of gestation both on antepartum and in intrapartum	Clinical Environment  Romano et al. (2016)	
11	 Di Tommaso et al. (2013)	Private Dataset	Ninety-seven traces of the FHR were selected among those collected between June and September 2009	Careggi University Hospital of Florence (Tuscany, Italy)	
12	 Chen et al. (2014)	Private Dataset	Sixty-two CTG searches with 20 to 30-minutes of sections collected from different pregnant women at the time of entry to the delivery room for labor pain	Mackay Memorial Hospital, a tertiary referral center	
13	 Arif (2015)	Private Dataset	Datasets consists of 2126 cardiotocograms	Maternity and Gynecology Clinic (Porto University Hospital in Portugal)	

Techniques of validation

In analyzing general results, researchers evaluated independent datasets using techniques. In addition, to evaluate the classification model proposed using validation techniques (Chinnasamy, Muthusamy & Gopal, 2013). To analyses the results of generalization, researchers evaluated an independent dataset by using a technique. Basically, validation techniques are used when the aim of the study is to predict and estimate the precision of a predictive model in practice (Chinnasamy, Muthusamy & Gopal, 2013). Table 3 illustrates these validation techniques used in the included articles of our survey.

In the Table 3, there are 6 validation techniques used in this review. Firstly, Cross validation was divided into 3 techniques, there are 10-fold cross validation, k-fold cross validation and 2-fold cross validation, and note that cross validation are the most widely used in this review paper. Other validation techniques are Kappa Statistics, Paired sample t-test, confidence interval, Bland-Altman approach and Mann–Whitney test.

Performance measure

The most common way to evaluate the used of various classifiers in our survey is the Evaluation using criteria. This method can assess the effectiveness of the classification methods. Classifier performance evaluation using various measurements such as confusion matrix, RMSE, selectivity, specificity, precision, accuracy, sensitivity, f-measure, quality index, gmean, MAE, AMAE, chi-square, spearman correlation, AUC, training time, testing time, misclassification error and confidence interval. Hence in this section all the above parameters are discussed. Table 4 shows the measurement criteria implemented in the reviewed articles.

Table 3 Model validation techniques used in the reviewed researches.

No	Name of Validation Technique	Reference	
1	Cross Validation (10 fold cross validation, k-fold cross validation and 2 fold stratified cross validation)	 Georgoulas et al. (2014), Gavrilis, Nikolakopoulos & Georgoulas (2015), Shah et al. (2015), Chamidah & Wasito (2015), Jyothi, Hiwale & Bhat (2016), Magenes et al. (2016), Warmerdam et al. (2016), Cömert & Kocamaz (2017b), Zhang & Zhao (2017), Georgieva et al. (2013), Xu et al. (2014), Yılmaz (2016), Cömert & Kocamaz (2017a), Kim, Yang & Lee (2017), Spilka et al. (2017), Georgoulas et al. (2017), Sahin & Subasi (2015), Chinnasamy, Muthusamy & Gopal (2013), Sundar, Chitradevi & Geetharamani (2014), Arif (2015)	
2	Confidence interval	 Das, Roy & Saha (2015b)	
3	Paired sample t-test	 Das, Roy & Saha (2015b), Ghi et al. (2016)	
4	Bland-Altman approach	 Das, Roy & Saha (2015b)	
5	Mann–Whitney test	 Ghi et al. (2016)	
6	Kappa Statistics	 Rei et al. (2015), Garabedian et al. (2017), Di Tommaso et al. (2013), Chen et al. (2014)	

Table 4 The performance measurement criteria implemented in the reviewed articles.

Refrefences	Reliability group								
	Confusion matrix																			
	TP %	TN%	FP%	FN%	RMSE	Selectivity	Specificity	Precision	Accuracy	Sensitivity	F-measure	Quality index	Gmean	MAE	AMAE	Chi-square	Spearman correlation	AUC	Training time	Testing time	Miscassification error	Confidence interval	
Tomas et al. (2013)	*								*														
Haweel & Bangash (2013)					*																		
Georgoulas et al. (2014)																*							
Rotariu et al. (2014)	*	*	*	*		*	*										*						
Gavrilis, Nikolakopoulos & Georgoulas (2015)								*		*													
Shah et al. (2015)								*		*	*												
Das, Roy & Saha (2015a)									*														
Chamidah & Wasito (2015)									*									*					
Cömert, Kocamaz & Güngör (2016)	*	*	*	*		*	*		*	*									*	*			
Jyothi, Hiwale & Bhat (2016)							*		*	*													
Magenes et al. (2016)	*	*	*	*					*														
Warmerdam et al. (2016)							*		*	*													
Cömert & Kocamaz (2017b)	*	*	*	*			*		*	*		*											
Georgoulas et al. (2017)									*				*	*	*								
Frigo & Giorgi (2017)					*									*									
Permanasari & Nurlayli (2017)																					*		
Nagendra et al. (2017)									*														
Zhang & Zhao (2017)	*	*	*	*			*	*	*	*	*	*						*					
Sahin & Subasi (2015)	*	*	*	*			*		*	*	*							*					
Ghi et al. (2016)																		*				*	
Rei et al. (2015)									*	*												*	
Romano et al. (2018)					*																		
Ocak (2013)									*														
Georgieva et al. (2013)	*	*	*	*			*			*													
Di Tommaso et al. (2013)																						*	
Czabański et al. (2013)	*	*	*	*			*		*	*		*											
Ocak & Ertunc (2013)									*														
Czabanski et al. (2015)												*											
Yılmaz (2016)	*	*	*	*					*														
Cömert & Kocamaz (2017a)							*			*													
Gamboa et al. (2017)							*			*													
Kim, Yang & Lee (2017)					*																		
Spilka et al. (2017)							*			*													
Chinnasamy, Muthusamy & Gopal (2013)								*	*	*	*												
Inbarani, Banu & Azar (2014)							*			*	*												
Sundar, Chitradevi & Geetharamani (2014)										*	*												
Arif (2015)								*	*	*	*												
Total	10	9	9	9	4	2	13	5	19	18	7	4	1	2	1	1	1	4	1	1	1	3	
%	20	18	18	18	8	4	26	10	38	36	14	8	2	4	2	2	2	8	2	2	2	6	

Various studies on feature extraction and cardiotocography classification are as presented in Table 4, as well as a full survey of different criteria and sub-criteria for the evaluation and benchmarking. Moreover, to evaluate the cardiotocography classification method, the criteria used are confusion matrix, RMSE, selectivity, specificity, precision, accuracy, sensitivity, f-measure, quality index, gmean, MAE, AMAE, chi-square, spearman correlation, AUC, training time, testing time, misclassification error and confidence interval. As clearly shown in Table 4, not all articles in this scope of review utilized all the above-mentioned criteria.

Motivation

From the articles that we reviewed, we found that researchers were motivated to develop and use automated methods to diagnose, feature extraction and classification of cardiotocography focused on improving early detection and rapid diagnosis, accuracy, guidelines, data sets, ONG experts, methods and techniques. This section describes the benefits obtained from the literature, mapped as benefits from the same group including quotations for each benefit for more discussion. A brief summary of the motivation for the feature extraction and cardiotocography classification is shown in Fig. 6.

Figure 6 Motivation categories of feature extraction and cardiotocography classification.

Motivation related to early detection and rapid diagnosis

In most cases oxygen insufficiency is normal, but the prolonged strain on the fetus for a longer period of period due to a week defense mechanism may lead to acidosis, as in Rotariu et al. (2014) the best way to prevent fetal brain damages or even fetal morality is achieved by early detection of fetal oxygen insufficiency is crucial. in Das, Roy & Saha (2015b), automatic interpretation systems are needed to describe the initial signs of hypoxia so as to avoid further compromise. During the intrapartum period, fetal distress causes continuous fetal oxygen deficiency resulting in perinatal morbidity and mortality, (Shah et al., 2015) and (Permanasari & Nurlayli, 2017) consistent monitoring and appropriate intervention are important to prevent maternal and fetal morbidity and mortality. CTG tracing computerized analysis is a major advance towards the solution to early identification of pre-natal pathology (Magenes et al., 2016). Early detection and prediction of pathological results can help reduce fetal morbidity and mortality worldwide as well as to prove whether the surgical intervention, such as cesarean section is needed as reported in Fergus, Selvaraj & Chalmers (2018)

Motivation related to diagnosis accuracy

Medical decision making relies heavily on automatic analysis of medical data, this greatly helps improve diagnosis and medical care (Das, Roy & Saha, 2015a). Cardiotocography is widely used by doctors to obtain detailed physiological information from fetuses and pregnant women as a technique for diagnosing fetal well-being (Zhang & Zhao, 2017). In article (Fergus, Selvaraj & Chalmers, 2018), automated computer analysis of cardiotocography signals showed strong evidence in diagnosing actual perinatal complications and predicted the onset of pathological results with far less variability and better accuracy. The advances in modern obstetric practice allowed many robust and reliable machine learning techniques to be utilized in classifying fetal heart rate signals (Cömert & Kocamaz, 2017a).

Motivation related to guidelines

In the field of cardhotocography, there are many guidelines that show how to determine FHR and fetal general health condition. In this observation, RCOG, DFRMT and to a lesser extent, the SOGC and NICHD classifications showed greater caution in defining “normal” patterns. This can contribute to higher levels of intervention, which leads to an increased incidence of cesarean section. On the other hand, classification with Parer & Ikeda found the highest level in the Reassuring pattern (Di Tommaso et al., 2013). Additionally, the 2015 FIGO intrapartum cardiotocography guidelines displayed good interobserver agreement, ease of usage that is felt high and the level of intervention that is moderate (Bhatia et al., 2017). CTG different guidelines such as FIGO, ACOG, NICHD, RCOG etc., there is a lot of variation in the expression of the final state, which varies from one guideline to another. Table 5 represents the frequency of use of the various research guidelines used in this study.

Table 5 Number of different guidelines occurrence in the study.

No.	FHR guidelines Types	Reference	
1	FIGO (International Federation of Obstetrics and Gynecology)	 Nagendra et al. (2017), Das, Roy & Saha (2015b), Cömert & Kocamaz (2017b), Gavrilis, Nikolakopoulos & Georgoulas (2015), Fergus, Selvaraj & Chalmers (2018), Czabański et al. (2013), Czabanski et al. (2015) Chudáček et al. (2014), Das, Roy & Saha (2015a), Kim, Yang & Lee (2017), Spilka et al. (2017), Romano et al. (2016), Wróbel et al. (2013), Bhatia et al. (2017), Rei et al. (2015), Garabedian et al. (2017), Nunes & Ayres-de Campos (2016)	
2	NICHD (the National Institute of Child Health Development)	 Das, Roy & Saha (2015b), Das, Roy & Saha (2015a), Romano et al. (2016), Rei et al. (2015), Di Tommaso et al. (2013), Gamboa et al. (2017), Garabedian et al. (2017)	
3	NICE (National Institute of Health and Care Excellence)	 Pinas & Chandraharan (2016), Das, Roy & Saha (2015a), Bhatia et al. (2017)	
4	ACOG (the American College of Obstetricians and Gynecologists)	 Das, Roy & Saha (2015b), Das, Roy & Saha (2015a)	
5	RCOG (the Royal College of Obstetricians and Gynecologists)	 Das, Roy & Saha (2015a), Ghi et al. (2016)	
6	CNGOF (the French College of Gynecology and Obstetrics)	 Garabedian et al. (2017)	
7	NICHHD (the National Institute of Child Health and Human Development)	 Nunes & Ayres-de Campos (2016)	

Motivation related to datasets

In this work (Arif, 2015), researchers used larger testing datasets for better analysis. The most important contribution of this research is the identification of significant features (only seven) upon feature selection process and further used for classification with classification accuracy comparable to original features.

Motivation related to obstetricians and gynecologist (ONG) experts

In order to overcome persistent inter and intra-observer variability, special research efforts are needed to incorporate physician domain knowledge into automatic decision systems (Georgoulas et al., 2014). In the work by (Shah et al., 2015) and (Chamidah & Wasito, 2015), inconsistencies in visual evaluation can be eliminated by developing clinical decision support systems. Cardiotocography data is useful for obstetricians in diagnosing fetal abnormalities and can be used to decide on medical intervention before continuous damage to the baby, but the interpretation of cardiotocography data carried out by obstetricians can never visually be objective (Sahin & Subasi, 2015). In this information era, the use of machine learning tools in medical diagnosis has increased gradually. This is mainly because the effectiveness of classification and recognition systems that have demonstrated able to improve in a great deal to help medical experts in diagnosing diseases (Sundar, Chitradevi & Geetharamani, 2014).

Motivation related to techniques and methods

in this section, the motivations of classifying techniques and methods implemented in the studies of our survey will be presented. Table 6 tabulates the motivation to adopt cardiotocography classification techniques and methods.

The various methods and techniques used in cardiotocography classification are presented in Table 6. Most of the articles surveyed in this study area expressed intention of achieving the best accuracy level and highest performance from the classification model. The researcher sought to increase the precision and the performance by offering a different classification or feature selection method. In most studies, the results showed good performance and high accuracy of the classification models based on the suggested method, showing that this area of research suggests many effective methods for feature extraction and cardiotocography classification.

Challenges

Over the past few years, there has been an increase in the field of feature extraction and cardiotocography classification, although there are still encounter difficulties and challenges in numerous important respects. Various examples of these challenges are diagnosis, parameters, guidelines, and many more. In Table 7 presents details about reference challenges.

Challenges related to classification algorithm

In Tomas et al. (2013), Generally, the challenge related to classification algorithms like (The Random Forest, Linear Discriminant Analysis (LDA) and REPTree) is based on the using of group classification. One of the commonly used and most promising statistical modeling methods is the Random Forest. The most important target is to reduce the error of the entire forest and the individual tree quality is not important. The error of the random forest relies on the correlation between the two trees and on the strength of the tree. The Random Forest evaluates which variables are essential in the classification and runs effectively on big databases. The Linear Discriminant Analysis (LDA) is a technique utilized for data classification. This method can catch the main difference between classes, maximizes the ratio between-class variance to the within-class variance, and discount irrelevant factors. The general purpose is to discover and analytically express boundary which would most discriminate between the groups. REPTree as a fast decision tree learner represents a set of machine learning algorithms for data mining and it has many tools like regression, clustering, data preprocessing, classification, visualization, and association rules. This technique sorts values for numeric attributes once and using reduced-error pruning. There are options available for the REPTree. For the three-algorithm implemented to classify CTG, the REPTree algorithm has a low accuracy mainly in the suspected pathological conditions for the small training sets compared to the other two algorithms. Three algorithms were used such as Random Forest, Linear Discriminant Analysis (LDA), and REPTree to classify CTG.

Table 6 Motivation to adopt cardiotocography classification techniques and methods.

	Methods & Techniques	Motivation	
1	Random Forest	Random forest is widely used in statistical modeling techniques and one of the most promising methods appears (Tomas et al., 2013). Random Forest has managed to increase classification accuracy and accomplished better generalization even for large databases in ensemble learning.	
2	Volterra Neural Network (VNN)	VNN has fast and uniform convergence. Simulation has demonstrated the efficiency of this techniques as proposed in electronic fetal monitoring (Haweel & Bangash, 2013).	
3	A Novel Software “CTG-OAS”	Magenes et al. (2016) has introduced a new software called CTG-OAS for a comprehensive analysis of CTG signals in this study. This software provides several tools for conducting reliable analysis. Also, important procedures regarding machine learning, such as feature extraction, pre-processing, classification and feature selection, are inherent in the software.	
4	Scattering Transform	Scattering transformation is proposed as a new tool for analyzing the variability of intrapartum fetal heart rate (FHR). This consists of a nonlinear extension of the underlying wavelet transformation, thereby maintaining its multiscale nature (Chudáček et al., 2014).	
5	Bagging Approach	In this study, a bagging approach combined with three traditional decision tree algorithms (random forest, Reduced Error Pruning Tree (REPTree) and J48) has been applied to identify normal and pathological fetal conditions using CTG data (Chudáček et al., 2014).	
6	Support Vector Machine (SVM)	SVM gives good accuracy (Chamidah & Wasito, 2015) and has showed high performance for binary classification with its ability to handle noisy data (Nagendra et al., 2017). SVM classification has become the most preferred techniques since its introduction, and has been successfully used in many medical decision support systems (Ocak, 2013).	
7	Artitifical Neural Network (ANN)	ANN is a practical tool to solve many complex nature signal processing problems, such as curve installation, pattern recognition and classification, grouping and analysis of dynamic time series (Cömert, Kocamaz & Güngör, 2016).	
8	Extreme Learning Machine (ELM)	ELM tends to provide good generalization performance with fast learning speed in many cases and can learn thousands of times faster than conventional learning (Cömert, Kocamaz & Güngör, 2016).	
9	K Nearest Neighbor	The classification method based on K Nearest Neighbor is presented for automatic classification of various uterine construction during labor (Jyothi, Hiwale & Bhat, 2016). K Nearest Neighbor is an example of a classification method with parameter independence (Sahin & Subasi, 2015).	
10	The Adaptive Boosting (AdaBoost)	Typical ensemble learning algorithms, this study proposed new design concepts and make great success in many different practical applications (Zhang & Zhao, 2017).	
11	Fuzzy Classification System	The possibility of assessing the efficient state of the fetus using the proposed fuzzy inference method (Czabański et al., 2013).	
12	Sparse Support Vector Machine (Sparse-SVM)	Permits to select a small number of relevant features and to achieve efficient fetal acidosis detection (Spilka et al., 2017).	

Table 7 Challenge categories of feature extraction and cardiotocography classification.

Challenge related to algorithm	•   The REPTree algorithm has low accuracy especially in the accuracy of suspected pathological status for small training sets.	
Challenge related to diagnosis	•  The knowledge and experience of the doctor largely influences accuracy;	
	•  Increasing cesarean delivery rates is one of the main reasons caused by a lack of information provided by cardiotocography;	
	• The available evidence about the accuracy and efficacy of this system is still limited.	
Challenge related to guidelines	• In the guidelines there is still lack of objective explanations for some features of fetal heart rate;	
	• The existing guidelines have deficiencies in terms of uniformity and uncertainty, therefore it is difficult to implement automatic systems;	
	• There is still lack of precision, leading to differences of opinion among medical practitioners;	
	• FIGO results poor specificity;	
	• guidelines have not become more simple or more objective;	
	• Differ in the terminology used;	
	• NICHD system was rapidly criticized by some investigators;	
	• The usefulness of NICHD system is under debate;	
	• no evidence of NICHD effectiveness.	
Challenge related to features	• Many different pattern in the gray zone;	
	• Baseline is the most basic feature of FHR;	
	• The missing value problem is another problem that needs to be resolved;	
	• Features extracted from histogram data are less important;	
	• very difficult to assess;	
	• Complex FHR patterns are assessed with eyes that are prone to errors, inconsistent and unreliable;	
	• The complexity of patterns describing the FHR variability makes the visual signal interpretation difficult and the accuracy of the analysis depends mostly on clinician’s knowledge and experience;	
	• The UA signal is often of poor quality.	
Challenge related to waveform	• The FHR waveform has a complex form;	
	• The FHR waveform is a source of a lot of information, only a small portion can be extracted by visual analysis;	
	• Heart rate signals often show complex and irregular fluctuations.	
Challenge related to dimensionality	• The high dimension of CTG data are the problem for classification computation;	
	• There is no guarantee that dimensions are higher and computing time is greater;	
	• K-SVM reduces the feature dimension, however features of other data with similar samples are not reduced.	
Challenge related to CTG interpretation	• Increased birth by caesarean section and less specific in detecting acidosis;	
	• Subjective interpretation and tedious technique;	
	• Poor specificity, not always possible;	
	• Unnecessary intervention;	
	• Poor positive predictive value;	
	• no standardization in the interpretation of the information;	
	• because the fetus is in the womb, several measurement problems arise;	
	• high complexity of signal patterns, which results in high levels of intra and interobserver variability;	
	• CTG has not proven its benefits in neonatal death and morbidity;	
	• Conventional visual CTG interpretation is limited.	
Challenge related to classification	• The FHR pattern classification still needs further improvement;	
	• Ignoring the suspect cases;	
	• Piquard classification was not applicable;	
	• The risk of false classification of pathological cases remains high;	
	• The predictive capacity of the existing methods remains inaccurate;	
	• Traditional unsupervised methods provide very poor accuracy in predicting different classes.	
Challenge related to time of diagnosis	• Training time and test time takes longer;	
	• SVM consumes a lot of computational time	
Challenge related to ONG experts	• Expertise is not always available, making CTG evaluation a difficult task;	
	• Interpretation of CTG data after visual analysis performed by obstetricians cannot be objective;	
	• the agreement between clinicians was moderate	
	• CTG recordings are analyzed by experts visually who make subjective interpretations and cannot be reproduced.	
Challenge related to technical challege	• The FHR record suffers from samples that are often invalid or lost, due to sensor artifacts or error functions;	
	• great inter- and intra-observer variability;	
	• Lack of patients identification still occurs in 8% of antepartum searches and 31% of intrapartum searches.	
Challenge related to dataset	• Data interpretation ambiguity;	
	• Lack of late acceleration in the dataset;	
	• could not be objective and reproducible;	
	• Outlier value problem.	
Challenge related to evaluation	• The classification uses a measure of performance evaluation, but it is not enough to decide for a vital case especially in a medical diagnosis;	
	• fetal hypoxia is only about 30% of the positive predictive value for intrapartum cases;	
	• There has been a effective increase in surgical birth rate and intrapartum cesarean section ;	
	• The standard definition of FHR variability (FHRV) and agreement on the methodology that will be used in its evaluation is still lacking;	
	• ANN is still not accepted as a valid tool	
	• One of the main disadvantages of Tocography is subjectivity in interpretation and has high frequency noise due to sudden movements during recording.	

Challenges related to diagnosis

Explanation of information provided by CTG is not in accordance with any standards, the lack of information provided by CTG is one of the main reasons for the increase in cesarean delivery (Shah et al., 2015). In Nunes & Ayres-de Campos (2016), computer analysis of fetal monitoring signals is proposed, but for now, available evidence of the accuracy and efficacy of this system is still limited. In the traditional mode of CTG interpretation by visual analysis widely practiced and less time consuming, its accuracy depends in large part on the knowledge and experience of the doctor (Das, Roy & Saha, 2015b).

Challenges related to guidelines

For the last 25 years the guidelines have not become more simpler or more objective, which could guarantee their wide application (Wróbel et al., 2013). Guidelines for interpretation have been proposed by several different organizations in the terminology used (Bhatia et al., 2017), these guidelines are not only lack uniformity, but also lack of objective explanation in some FHR features (Das, Roy & Saha, 2015b) and also uncertain, therefore, it is difficult to implement in an automated system (Das, Roy & Saha, 2015a). In addition, FIGO or NICHD guidelines are based solely on experimental observations, they lack precision, poor specificity (Chudáček et al., 2014) lead to differences of opinion among medical practitioners (Das, Roy & Saha, 2015a). NICHD systems was rapidly criticized by some investigators, the usefulness of NICHD system is under debate and still no evidence of NICHD effectiveness (Gamboa et al., 2017).

Challenges related to features

In clinical practice, complicated FHR patterns are assessed visually (using eyes), which are prone to error, inconsistent and unreliable (Georgieva et al., 2013), and many different pattern in gray zone (Das, Roy & Saha, 2015b). Not only in FHR pattern, the UA signal is often of poor quality (Warmerdam et al., 2018). Baseline is the most basic feature of the FHR because all the other features are directly dependent on it (Das, Roy & Saha, 2015a). In addition to the baseline, the complexity of the patterns that describe FHR variability makes interpretation of visual signals difficult and the accuracy of the analysis depends largely on the knowledge and experience of the doctor (Czabanski et al., 2015). Another drawback is the missing value problem and this that needs to be resolved (Chamidah & Wasito, 2015). Based on ratings, features extracted from histogram data were less significant compared to features extracted from the Fetal Heart Rate and readings of the Uterine Construction time series (Nagendra et al., 2017).

Challenges related to waveform

The FHR waveform is a source of numerous information, only a small portion can be extracted by visual analysis (Das, Roy & Saha, 2015a), but the FHR waveform is complex (Das, Roy & Saha, 2015b). in Warmerdam et al. (2016), Heart rate signals often showed complex and irregular fluctuations that cannot be explained by spectral analysis.

Dimensionality challenges

The high dimension relating to data input poses a problem in classification. If the dimensions are high, the process of training machine learning will require large CPU time. But there is no guarantee that higher dimensions and greater computation time will also increase accuracy (Chamidah & Wasito, 2015). As reported in Chamidah & Wasito (2015), K-SVM reduced the feature dimension, but failed to reduce the data with the same sample.

Challenges related to CTG interpretation

CTG has not proven yet its benefits in neonatal death and morbidity. This lack of benefits may reflect damage to the fetus during the procedure (Bhatia et al., 2017). Conventional CTG visual interpretation is limited, and many previous studies have documented high intra-observer and inter-observer variations (Chen et al., 2014). The main problem in interpreting the correct record is the high complexity of the signal pattern, which results in a high level of intra and inter observer variability (Czabański et al., 2013), subjective interpretation and tedious technique (Rotariu et al., 2014). EFM has a poor positive predictive value, while a high false positive rate is used to detect fetal hypoxia in the intrapartum period (Cömert & Kocamaz, 2017b). Besides that, poor specificity of CTG leads to unnecessary interventions (Warmerdam et al., 2016). Moreover, since the fetus is in the womb, several measurement problems have arisen (Romano et al., 2018). The consequence of various interpretations of CTG has increased cesarean delivery and less specific in detecting acidosis (Chamidah & Wasito, 2015). Also, there is no standardization in the interpretation of the information which was provided by CTG (Permanasari & Nurlayli, 2017).

Challenges related to classification

Artificial Neural Networks, Fuzzy Systems, Genetic Algorithms and Supporting Vector Machines for prediction of fetal conditions have been developed and tested, but these classifiers are only applied to normal and pathological cases, ignoring suspected cases. (Nagendra et al., 2017). In four fetuses with piqued classification, acidemia could not be confirmed because the pattern of fetal heart rate (tachycardia, reduced to no variability or slow deceleration with normal baseline) is not included in one of the six categories (Ghi et al., 2016). The risk of false classification in pathological cases remains high in Indonesia (Czabański et al., 2013). Even few decades after the introduction of cardiotocography into clinical practice, the predictive capacity of the existing methods remains inaccurate (Chinnasamy, Muthusamy & Gopal, 2013). Conversely, traditional grouping methods can identify normal CTG patterns, they are incapable of Suspicious and Pathological patterns, so, traditional unattended methods provide very poor accuracy in predicting different classes (Sundar, Chitradevi & Geetharamani, 2014). Although many obstetricians use CTG, the classification of the FHR pattern still needs further improvement (Cömert, Kocamaz & Güngör, 2016).

Challenges related to time of diagnosis

Training time and testing time take longer and there is also no significant increase in the success of network classification of more than 25 hidden layer sizes (Cömert, Kocamaz & Güngör, 2016). In Spilka et al. (2017), the SVM method required very long running time. In addition, another challenge was due to lower classification or prediction accuracy if data involved has complex characteristics such as noise, non-linearity, non-stationery and others.

Challenges related to ONG experts

Interpretation of CTG data after visual analysis performed by obstetricians must be ascertain Sahin & Subasi (2015), not based on subjective interpretation and cannot be reproduced (Ocak, 2013). When comparing 4 FHR classification, our study indicates that the agreement between clinicians was moderate no matter what are the proposed classification and even with a 5-tier system (Garabedian et al., 2017). Besides that, Expertise is not always available, making CTG evaluation a difficult task (Georgoulas et al., 2017).

Challenges related to technical matters

The FHR record suffered from samples that are often invalid or lost, due to sensor artifacts or error functions (Frigo & Giorgi, 2017). When CTG techniques are introduced into clinical practice, recordings are interpreted by gynecologists and midwives through visual inspection, with clear consequences of large inter-and intra-observer variability, which leads to unreliable conclusions about fetal morbidity (Zhang & Zhao, 2017). In Nunes & Ayres-de Campos (2016), the lack of patients identification still occurs in 8% of antepartum searches and 31% of intrapartum searches.

Challenges related to datasets

Interpretation of data through visual analysis performed by obstetricians cannot be objective and can be reproduced (Zhang & Zhao, 2017). In addition, the ambiguity of data interpretation also depends on the quality of the data being measured (Frigo & Giorgi, 2017). The lack of late acceleration in the dataset is a big problem, because slow acceleration followed by short-term variability is a high indicator of fetal acidosis (Nagendra et al., 2017). In addition, outlier value problems are other important problems that need to be solved. In Zhang & Zhao (2017), we use datasets that do not has outlier values.

Challenges related to evaluation

As we know, classification uses performance measure for evaluation purpose, but it is still not enough to decide for a vital case especially for medical diagnosis (Sahin & Subasi, 2015). Although to predict fetal hypoxia during pregnancies, CTG was basically developed as a screening tool, the positive predictive value for intrapartum fetal hypoxia is only about 30% (Pinas & Chandraharan, 2016). There has been a substantial increase in operative vaginal delivery rates and intrapartum cesarean section (Pinas & Chandraharan, 2016). Although it is clinically important and widespread use of fetal monitoring, the standard definition of FHR variability (FHRV) and agreement on the methodology to be used in its evaluation are lacking (Romano et al., 2018). ANN is still not accepted as a valid tool for classification purpose mainly because it is difficult to explain how a diagnosis is achieved; since, ANN is considered as a nonlinear black box (Georgieva et al., 2013). On the other hand, one of the main disadvantages of Tocography is subjectivity in interpretation and it has high frequency noise due to sudden movements during recording (Jyothi, Hiwale & Bhat, 2016).

Recommendation

This section offers several important recommendations for literature specifically recommended categories of cardiotocography classification and feature extraction as described in Fig. 7.

Figure 7 Recommended categories of feature extraction and cardiotocography classification.

Recommendations to developers and researchers

Recommendations for developers and researchers offered in this section of the study and most recommendations here are related to the performance and accuracy of feature extraction and cardiotocography classification.

Accuracy Recommendation.

From the articles reviewed, there is a possibility of increasing the classification accuracy through several recommendations. These recommendations are associated with accuracy that focuses on informative features, estimated parameters, and classification models.

CTG diagnosis accuracy depends on the analysis of characteristic FHRs and UCs (Chen et al., 2014). FHR, recorded either by ultrasound Doppler probe or by a scalp electrode, contains various artifacts, therefore it is necessary to preprocess the FHR signal prior to feature extraction (Georgoulas et al., 2014). In Chamidah & Wasito (2015), feature extraction and selection is performed to obtain the hidden pattern of normal, irregular and pathologic fetal state separately using K-Mean algorithm. The process of identification is based four parameters (baseline, variability, acceleration and deceleration) and these need to be carefully evaluated and extreme care is needed to interpret data at the boundaries region (Das, Roy & Saha, 2015a). A larger dataset and other classifiers are required to gain insight in which combination of HRV features is the most informative (Warmerdam et al., 2016). Not only additional data is essential for parameter estimation, the opinion of more physicians with varied level of experience is required too (Das, Roy & Saha, 2015a). Additionally as reported in this study, seven features (AC, DS, DP, ASTV, MSTV, ALTV and Mean) are considered more appropriate for CTG data analysis (Shah et al., 2015). To improve consistency in CTG practice, classification systems must be universally standardized to enable the development and dissemination of educational tools (Bhatia et al., 2017). Besides that, inclusion of clinical information such as gestational age or maternal temperature might also enhance the classification (Warmerdam et al., 2016) and to avoid the bias of interpretation, different classification techniques have been proposed (Garabedian et al., 2017). Unclassifiable CTG trace encountered during the second stage of labor should not be regarded simply as a technical artefact but should always alert the clinician about an underlying fetal hypoxia and prompt further testing to assess the fetal status more accurately including internal scalp monitoring or fetal blood sampling (Ghi et al., 2016).

Performance recommendation.

It is indeed vital to include more information during the process of labor especially the time-series data, that should be considered to be integrated into the classifier for better performance. Moreover, there are also additional clinical parameters that need to be investigated, such as gestation, oxytocin augmentation, maternal infection, and many more (Xu et al., 2014). Besides that, examining the classification performance over time (Warmerdam et al., 2016) and excluding some features to optimize computational time and to achieve better performance (Zhang & Zhao, 2017), can also be done using performance evaluation tools (Sahin & Subasi, 2015). As reported in Das, Roy & Saha (2015a), width of the Confidence Interval suggested that more data need to be included in the sample. As for the propose technique by Sundar et al., hybrid models using statistical (Sundar, Chitradevi & Geetharamani, 2014) and all machine learning techniques produced rather well performances, ANN was shown superior to others (Cömert & Kocamaz, 2017a). As a future direction, neural networks can be used to tune the parameters and to achieve better performance (Inbarani, Banu & Azar, 2014). A fuzzy output of the fetal state with a membership function over different periods of time can be suitable for a real-time clinical decision support system (Nagendra et al., 2017).

Recommendations to users

Before starting any CTG recording, it is mandatory to check the maternal pulse to avoid erroneous recording of maternal heart rate as fetal (Pinas & Chandraharan, 2016). gynecologists and clinicians need to distinguish the physiology behind FHR changes and then how to respond to them accordingly to each individual cases, instead of purely relying on guidelines for management (Pinas & Chandraharan, 2016). Guidelines help practitioners, whose task is to interpret CTG traces, to understand different patterns of FHR in a more straightforward manner (Cömert & Kocamaz, 2017b). Clinicians must understand the types of intrapartum fetal hypoxia and fetal reserve is also vital to optimize outcomes (Pinas & Chandraharan, 2016), and clinicians must be able to provide better diagnosis of the problem because it is extremely important to avoid unnecessary interventions (Garabedian et al., 2017).

Limitation

There is some limitation from the literatures, for instance small sample size might mean the study was inadequately powered to make reliable extrapolations, inexistence of universally accepted guidelines. Some other limitations in this study include objective CTG definition that resulted in limited effectiveness of CTG monitoring, the use of cardiotocography for considerable intra and inter observer variability, low classification accuracy, improvement via hybrid models using statistical and machine learning techniques, failed to analyze the relationship between the FHR patterns and neonatal outcomes as well as not informing mothers to count fetal movements during the intrapartum period and missing information on fetal movement.

Conclusion

As a conclusion, interpretation using cardiotocography is one of the most widely used methods in Electronic Fetal Monitoring (EFM). Research in this field is still ongoing, even though the description and relevant boundaries are still ambiguous. Hence it is crucial to gain understanding and insight into this research. This systematic review aimed to contribute to the research area through survey and classification of related research efforts. Research efforts in this study area are categorized into four namely: proposed methods, system development, surveys and reviews followed by evaluation and comparison of studies. Significant information was gathered by intensive reading, summarizing and analyzing various previous studies specifically that are related to, benefits and motivations, recommendations related to feature extraction and cardiotocography classification as well as measurement of various performance along with diverse datasets that have been used in previous researches. Further, challenges in this field of research have been identified and recommendations to overcome as well as method to solve the limitations and drawbacks are also provided. As compared to some other developing technologies; most studies aimed to focus on the functional qualities of the technology and consider the non-functional aspects as lower priorities. Users usutilize the technology that exists once the technology is made available. Therefore, to ensure that this research is in line with the current technology, researchers must focus on new technologies, such as the use of applications that can be employed using smartphones. To the extent of our knowledge, research on feature extraction and cardiotocography classification using smartphone-based applications has not been done, so this is an important research line that are needed with the hope that it may intersect with several other technological scientific pathways.

Additional Information and Declarations

Competing Interests

Author Contributions

Data Availability

The authors declare there are no competing interests.

Shahad Al-yousif, Ariep Jaenul and Wisam Al-Dayyeni conceived and designed the experiments, performed the experiments, analyzed the data, prepared figures and/or tables, authored or reviewed drafts of the paper, and approved the final draft.

Ah Alamoodi and Nooritawati Md Tahir conceived and designed the experiments, performed the experiments, authored or reviewed drafts of the paper, and approved the final draft.

IA Najm conceived and designed the experiments, analyzed the data, authored or reviewed drafts of the paper, and approved the final draft.

Ali Amer Ahmed Alrawi conceived and designed the experiments, prepared figures and/or tables, authored or reviewed drafts of the paper, and approved the final draft.

Zafer Cömert performed the experiments, analyzed the data, authored or reviewed drafts of the paper, and approved the final draft.

Nael A. Al-shareefi performed the experiments, authored or reviewed drafts of the paper, and approved the final draft.

Abbadullah H. Saleh conceived and designed the experiments, performed the experiments, prepared figures and/or tables, authored or reviewed drafts of the paper, and approved the final draft.

The following information was supplied regarding data availability:

This is a review article. No code or raw data were used or presented in this work.

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
