# Peer review of "A systematic review of automated pre-processing, feature extraction and classification of cardiotocography"

_PeerJ Computer Science, doi:10.7717/peerj-cs.452_

## Round 0.1 · original submission · Major Revisions

The paper clearly needs improvement. Please make all the changes based on the reviewer comments/feedback so that the paper quality will be better. You can use multiple tables to compare all the papers based on different criteria, along with textual comments. I would suggest you read review/survey articles in good journals, to see how they are written.

Reviewer 1 ·

Basic reporting

The manuscript presents a systematic review on CTG.The techniques,datasets , limitations and recommendations are presented using 50 articles.
Overall English language needs improvement.
A detailed review with more techniques might have been included.

Experimental design

.A detailed inclusion of classification techniques can be done.

Validity of the findings

Tables and figure names are inaccurate.
Table 2 and 3 shows the same title.
Review of Feature extraction techniques should be extensive.

·

Basic reporting

The article is written in English, the text is technically correct in general. However, it contains some unsuccessful expressions and errors, for example:
• Line 93-94: If deceleration is perceived with contraction, possibility indicates with hypoxia - apparently meant it probably indicates hypoxia.
• Line 99-100: However, the interpretations made by humans are inconsistent namely about the traces and variability of inter and intra high observer - it is not clear what the authors wanted to say.
• Similar examples are found in lines 67, 85, 92, 107, 108, 131, 179-180 etc.
The article is well structured. The authors should be commended for the successful nomenclature and content of the tables, which are very helpful in understanding the complex structure of the material.
The article is within the scope of the journal.
As far as I know, there have never been such full reviews on this topic before, so the proposed article is of great interest and value.
The Introduction adequately introduces the subject. However, since the article is interdisciplinary, it should be explicitly indicated who exactly it is addressed to - IT specialists, obstetricians, or both.

Experimental design

The survey methodology is consistent with a comprehensive, unbiased coverage of the subject. The review is logically organized, sources are adequately cited.
Authors use two types of classification of the reviewed articles. The second type of classification (Datasets, Techniques of Validation, Performance Measure, Motivation, Challenges, Recommendation) is quite clear and logically comes from the structure of the reviewed articles.
As for the first classification (the authors call it either taxonomy or stages), it is not clearly indicated. For example, the abstract (lines 41-44) says: “First stage discussed the proposed method which presented steps and algorithms in the pre-processing stage, feature extraction and classification as well as their use in CTG (20/50 papers). The second stage included the development of system specifically on automatic feature extraction and CTG classification (7/50 papers).” These expressions are very similar to each other and are not very suitable for taxonomy.
Moreover, lines 208-209 for the same articles (20/50) provide a different basis for the same classification: “The general context of all articles (20/50) in this category is about achieving high performance or increased in accuracy of CTG”.
Authors should be encouraged to define the basis for classification used more carefully and reproduce it similarly in all sections of their review.

Validity of the findings

Conclusions are well stated.
The fragment stating the meaning of the taxonomy of articles (lines 322-351) appears to be the authors' personal reasoning and does not follow from the articles reviewed. According to the journal's requirements (“Speculation is welcome, but should be identified as such”), this fragment should be labeled as speculation.
Essentially, the main conclusions of the review are contained in tables developed by the authors, which provide a comprehensive picture of the state of research in the subject area of cardiotocography. Such clear and compact presentation of the material should be really welcomed.

Additional comments

The article is of undoubted interest and after the elimination of pointed remarks can be recommended for publication in the journal

Reviewer 3 ·

Basic reporting

This article reports the work that aimed to provide a systematic review to describe the achievements made by the literature from the years 2013-2018, summarizing findings that have been found by previous researchers in feature extraction and CTG classification, to determine criteria and evaluation methods to the taxonomies of the proposed literature in the CTG field and to distinguish aspects from relevant research in the field of CTG.

Experimental design

The study design is sound and detailed.

Validity of the findings

The findings are well summarized. However, I found it quite shallow how the authors reported those findings. For instance, I expected some details on what are the features mostly used, what are the best-reached performance metrics. Currently, the article reads like a collection of the existing research, however, it can't act as a stand-alone for an adequate literature review of the topic.

Additional comments

Section 2.3 is redundant
Why 5 years?

Section 3 expected details on the best features

Placement of figures and tables could be enhanced

4.4.3. Motivation related to guidelines missing details

4.5.1 shallow

---

## Round 0.2 · accepted · Accept

The paper can be accepted as per the reviews done.

·

Basic reporting

The revised article meets all the requirements of the journal

Experimental design

The revised article meets all the requirements of the journal

Validity of the findings

The revised article meets all the requirements of the journal

Additional comments

The revised article meets all the requirements of the journal

Reviewer 3 ·

Basic reporting

This article reports the work that aimed to provide a systematic review to describe the achievements made by the literature from the years 2013-2018, summarizing findings that have been found by previous researchers in feature extraction and CTG classification, to determine criteria and evaluation methods to the taxonomies of the proposed literature in the CTG field and to distinguish aspects from relevant research in the field of CTG.

Experimental design

The study design is sound and detailed from the initial submission

Validity of the findings

I laud the authors for the rebuttal and the revised manuscript.
They managed to answer all my concerns and I think the submission is ready for acceptance.